# Non-Fasting Hypertriglyceridemia Burden as a Residual Risk of the Progression of Carotid Artery Stenosis

**DOI:** 10.3390/ijms23169197

**Published:** 2022-08-16

**Authors:** Yoichi Miura, Ryuta Yasuda, Naoki Toma, Hidenori Suzuki

**Affiliations:** Department of Neurosurgery, Mie University Graduate School of Medicine, 2-174 Edobashi, Tsu 514-8507, Mie, Japan

**Keywords:** atherosclerosis, carotid artery stenosis progression, non-fasting triglyceride

## Abstract

The relationships between repeated non-fasting triglyceride (TG) measurements and carotid stenosis progression during follow-ups have never been investigated. In 111 consecutive carotid arteries of 88 patients with ≥50% atherosclerotic stenosis on at least one side, who had ≥3 blood samples taken during ≥one-year follow-ups, clinical variables were compared between carotid arteries with and without subsequent stenosis progression. To evaluate non-fasting TG burden, a new parameter area [TG ≥ 175] was calculated by integrating non-fasting TG values ≥ 175 mg/dL (i.e., TG values minus 175) with the measurement intervals (year). Carotid stenosis progression occurred in 22 arteries (19.8%) during the mean follow-up period of 1185 days. Younger age, symptomatic stenosis, higher mean values of TG during follow-ups, the area [TG ≥ 175], mean TG values ≥ 175 mg/dL and maximum TG values ≥175 mg/dL were significant factors related to the progression on univariate analyses. The cut-off value of the area [TG ≥ 175] to discriminate carotid stenosis progression was 6.35 year-mg/dL. Multivariate analyses demonstrated that symptomatic stenosis and the area [TG ≥ 175] ≥ 6.35 year-mg/dL were independently related to carotid stenosis progression. In conclusion, the area [TG ≥ 175] was an independent risk factor for carotid stenosis progression, and this study suggests the importance to continuously control non-fasting TG levels < 175 mg/dL during follow-ups to prevent carotid stenosis progression.

## 1. Introduction

Although many types of intervention have been explored to treat dyslipidemia, particularly low-density lipoprotein cholesterol (LDL-C), which is an established risk factor for the development of cardiovascular and cerebrovascular diseases, the diseases have not yet been eradicated. In this context, triglyceride (TG) is attracting more attention as a residual risk factor of atherosclerotic diseases. Recent studies reported that elevated non-fasting TG levels had a marked impact on cardiovascular disease [1,2,3] and that non-fasting TG was a better predictor of the development of cardiovascular disease than fasting TG [4,5]. On the other hand, the relationships between non-fasting TG levels and the progression of carotid artery stenosis remain unclear. Recently, the authors reported that non-fasting hypertriglyceridemia at the initial diagnosis of ≥50% carotid artery stenosis and at the start of follow-up was an independent predictor of the subsequent carotid stenosis progression, when fasting serum LDL-C levels were controlled under 140 mg/dL [6]. In the study, it was suggested that more strict control of non-fasting TG levels may be necessary for more severe carotid stenosis to prevent the progression [6].

It is very useful in clinical practice if non-fasting serum TG values can effectively assess a risk of carotid stenosis progression, because postprandial TG measurements would allow clinicians to easily manage TG levels with a minimum of patients’ discomfort. However, there is concern that postprandial TG measurements may vary considerably due to the influence of meals and may be unreliable as an index. Since no study has investigated the relationships between changes or fluctuations in non-fasting TG levels during follow-ups and progression of carotid artery stenosis so far, the present study aimed to examine changes in non-fasting TG levels over time and to find a better influencer related to non-fasting TG that may independently aggravate carotid artery stenosis.

## 2. Results

The analyzed 111 carotid arteries in eighty-eight patients (eighty-three males and five females) had an average age of 70.6 years (range, 45 to 84). During a mean follow-up period of 1185 days, 22 of 111 carotid stenosis (19.8%) suffered stenosis progression: the mean degree of carotid stenosis progressed from 32.2% (range, 0 to 70) at baseline to 72.9% (range, 40 to 100) at the last follow-up on ultrasonography. Among the twenty-two progressively narrowed arteries, three caused ipsilateral ischemic stroke during the follow-up period, and eleven and two underwent CAS and CEA, respectively. The study included two symptomatic carotid stenosis of ≥50%, for which patients consented to neither CAS nor CEA at the initial DSA diagnosis: the carotid artery with symptomatic 60% stenosis progressed to 80% stenosis without ischemic stroke in 390 days and then was treated with CAS, while the other carotid artery with symptomatic 74% stenosis showed no significant progression of stenosis during the 600-day follow-up.

### 2.1. Comparison of Baseline Clinical Characteristics between the Progression and Non-Progression Groups

Baseline clinical characteristics of 111 carotid arteries were shown and compared between the progression and non-progression groups in Table 1. Sex, body mass index, baseline degree of carotid stenosis, SIR of a carotid plaque, past medical history, medical treatment including statins, and follow-up periods were not significantly different between the two groups. Younger age and symptomatic carotid stenosis were significantly related to the stenosis progression compared with the non-progression group. As to the baseline laboratory data, non-fasting serum values of total cholesterol, HDL-C, non-HDL-C, LDL-C, glucose and hemoglobin A1C were not significantly different between the two groups, but non-fasting TG values and the incidence of non-fasting hypertriglyceridemia were significantly higher in the progression group compared with the non-progression group.

### 2.2. Non-Fasting TG during the Follow-Up Period

To investigate effects of fluctuations in non-fasting TG values on carotid stenosis progression, the maximum, minimum and mean values of non-fasting TG, and the area [TG ≥ 175] were compared through the follow-up period (Table 2; all the TG measurement data are shown in Appendix A). The maximum and minimum values of TGs were not significantly different between the 2 groups, but the mean of all measured TG values and the median of the area [TG ≥ 175] were significantly higher in the progression group. The incidence of the maximum and mean TG levels ≥ 175 mg/dL was also significantly higher in the progression group compared with the non-progression group, although that of the minimum TG levels ≥ 175 mg/dL was similar. Among these non-fasting TG-related variables, the area [TG ≥ 175] had the lowest *p* value (Table 2).

The ROC curve analysis revealed that a cut-off value of the area [TG ≥ 175] to discriminate carotid stenosis progression was 6.35 year-mg/dL with the AUC of 0.691, a sensitivity of 0.682 and a specificity of 0.652 (Figure 1). The incidence of the area [TG ≥ 175] ≥ 6.35 year-mg/dL was also significantly higher in the progression group with the smallest *p* value and the highest OR among the non-fasting TG-related variables (Table 2).

### 2.3. Multivariate Analyses to Determine Variables Related to Carotid Stenosis Progression

Statistically significant clinical characteristics and non-fasting TG-related variables on univariate analyses were age, symptomatic carotid stenosis, mean TG values during follow-ups, incidences of the mean TG levels ≥ 175 mg/dL and maximum TG levels ≥ 175 mg/dL, area [TG ≥ 175], and the incidence of area [TG ≥ 175] ≥6.35 year-mg/dL (Table 1 and Table 2). Among them, age and non-fasting TG-related variables were significantly intercorrelated, and the incidence of the area [TG ≥ 175] ≥ 6.35 year-mg/dL had the smallest *p* value on univariate analyses. Symptomatic carotid stenosis was not significantly correlated with non-fasting TG-related variables including the incidence of area [TG ≥ 175] ≥ 6.35 year-mg/dL (r = 0.04). Thus, symptomatic carotid stenosis and the area [TG ≥ 175] ≥ 6.35 year-mg/dL were used as candidate variables for multivariate logistic regression analyses. Logistic regression analyses indicated that symptomatic carotid stenosis and the area [TG ≥ 175] ≥ 6.35 year-mg/dL were independent factors associated with carotid stenosis progression (OR = 7.167, 95% CI = 2.168–23.700, *p* = 0.001; OR = 4.827, 95% CI = 1.626–14.329, *p* = 0.005, respectively; Table 3).

## 3. Discussion

In the present study, it was found that the area [TG ≥ 175], cumulative non-fasting TG values more than 175 mg/dL during the follow-up period were a superior and independent risk factor of carotid stenosis progression over the mean of non-fasting TG values during the follow-up period. Surprisingly, the maximum and minimum values of non-fasting TG were not helpful to discriminate carotid stenosis progression. The findings suggest that longer exposure of carotid plaques to ≥175 mg/dL non-fasting TG is an important contributor to progressive atherosclerosis, rather than a transient high value of non-fasting TG. It is also suggested that a transient low value of non-fasting TG is not protective for carotid stenosis progression. Thus, it may be pivotal to control non-fasting TG stably to 175 mg/dL or less to prevent carotid stenosis progression.

It is well known that advanced age itself is related to pathogenic mechanisms of the development of carotid artery atherosclerosis in multiple ways [7]. In the present study, however, patients in the carotid stenosis progression group were significantly younger than those in the non-progression group on univariate analyses. The discrepancy can be explained by the finding that age was significantly correlated with non-fasting TG values. It has been documented that lifestyle factors including acute and habitual diet influence non-fasting TG, and higher TG values were sometimes reported in younger patients [8,9]. The findings in the present study probably reflect the results of younger patients preferring diets that raise TG values. Although symptomatic carotid stenosis was also an independent risk factor for the progression in this study, the finding may reflect the importance of local factors such as inflammation [10], which cannot be shown by degree of stenosis and SIR of plaques.

Elevated non-fasting TG levels are known to reflect high levels of remnant lipoproteins from chylomicrons and very low-density lipoproteins (VLDLs), and therefore can be a strong predictor of cardiovascular and cerebrovascular events [1,2,3,11]. Iso et al. reported that elevated non-fasting TG levels at medical examinations were associated with subsequent occurrence of myocardial infarction, angina pectoris, and sudden death, in a prospective study consisting of 11,068 Japanese people [1]. Nordestgaard et al. demonstrated in the Copenhagen City Heart Study that non-fasting TG levels correlated with the incidence of ischemic heart disease and ischemic stroke, leading to a joint consensus statement from the EAS and the European Federation of Clinical Chemistry and Laboratory Medicine that postprandial data are enough to assess a cardiovascular risk [11]. The statement sets the upper reference value of postprandial TG to 175 mg/dL, which is 25 mg/dL higher than the fasting reference TG value of 150 mg/dL [11]. Serum TG levels are generally increased for six to eight hours after a standard diet [6,12], and it may be feared that TG levels can be increased more than the estimated 25 mg/dL variance when a patient consumes a high-fat meal prior to blood sampling, leading to the misrepresentation of the clinical significance. Therefore, serum TG values have been regularly measured exclusively after eight to twelve hours of fasting [11]. However, it has been also reported that the effect of diet on TG levels is not so problematic without the association of dyslipidemia, and that a non-fasting TG value may be much higher than fasting TG values of healthy subjects only in case of the association of dyslipidemia or slow lipid particle clearance after diet [11,13]. In addition, most individuals are under a non-fasting state for most of the day, and fasting levels of TG may not reflect daily TG values. The present study thus assessed the effects of non-fasting TG levels on carotid stenosis progression, and adopted 175 mg/dL as the upper limit of the normal value of non-fasting TG.

Recently, the level of non-fasting TG is getting more attention as an atherosclerosis promoting factor, and recent guidelines recommend using non-fasting values rather than fasting ones for the screening and management of TGs [11,14,15]. However, the relationships between non-fasting TG levels and the progression of carotid atherosclerosis are not well established. Mori et al. reported the relationships between postprandial remnant-like particle TG levels and carotid intima-media thickness (IMT) in 68 patients with type-2 diabetes mellitus [16]. They divided type-2 diabetic patients using premeal and 2-hour postprandial TG levels into the normo-triglyceridemia, postprandial hypertriglyceridemia and fasting hypertriglyceridemia groups, and showed that the IMT values were significantly higher in the fasting as well as postprandial hypertriglyceridemia groups compared with the normo-triglyceridemia group, suggesting that delayed TG metabolism leading to the retention of remnants was closely associated with atherosclerosis [16]. Teno et al. found that non-fasting TG values were the greatest influencer for carotid IMT among fasting and non-fasting values of glucose, total cholesterol, LDL-C, HDL-C and TG in 61 Japanese patients with type-2 diabetes mellitus [17]. Idei et al. also revealed that a higher mean value of non-fasting TG during a 1-year period was more frequently associated with the appearance of carotid plaques and was a superior predictor to either the mean and one-point fasting TG values or one-point non-fasting TG ones in 115 patients with type-2 diabetes mellitus [18]. As to carotid artery stenosis, our recent study showed that elevated non-fasting TG level at the first visit to neurosurgical clinic to evaluate ≥50% carotid artery stenosis was an independent risk factor of subsequent progression of the disease in 96 patients under good control of fasting LDL-C levels [6]. Additionally, the study reported that a cut-off value of non-fasting TG to discriminate carotid stenosis progression was 169.5 mg/dL for less severe-side carotid arteries with the baseline stenosis of <50%, and 154.5 mg/dL for worse-side carotid stenosis of ≥50%, suggesting that more strict control of non-fasting TG was necessary to prevent the stenosis progression for a higher degree of carotid artery stenosis [6]. To our knowledge, however, no study has examined whether intra-individual variability of non-fasting TG levels during a follow-up period is involved in carotid stenosis progression. In the present study, the initial TG value was measured at the first DSA diagnosis of at least one-side significant carotid stenosis, maybe when the carotid artery stenosis had progressed or was first noticed, and it was confirmed that the initial one-point non-fasting TG value was significantly higher in the subsequent stenosis progression group. This may indicate that non-fasting TG is elevated during the phase of plaque growth. However, a one-time non-fasting TG measurement may underestimate the risk of carotid stenosis progression, because the mean values of maximum and minimum non-fasting TG as well as the incidence of the minimum TG levels ≥ 175 mg/dL were not significantly different between the progression and non-progression groups. On the other hand, higher mean non-fasting TG values of all measurements during the follow-up period, higher incidences of mean TG values ≥ 175 mg/dL or maximum TG values ≥ 175 mg/dL, and higher values of the area [TG ≥ 175] were associated with carotid stenosis progression on univariate analyses, and the area [TG ≥ 175] ≥ 6.35 year-mg/dL was found to be an independent predictor of the disease progression on multivariate analyses. This indicates the importance of continuous monitoring of non-fasting TG levels to prevent the progression of carotid artery stenosis. Further clinical investigations are needed to confirm the significance of controlling non-fasting TG values below 175 mg/dL constantly during follow-ups, and to clarify if TG lowering medications prevent carotid stenosis progression.

The impact of elevated non-fasting TG levels on carotid artery stenosis can be explained by the following mechanisms [19,20]. In the postprandial state, chylomicrons are secreted from the intestines into the blood, and the resulting increased chylomicron remnants lead to the production of TG [20]. In addition, under dyslipidemia with insulin-resistant states (obesity, metabolic syndrome and type-2 diabetes mellitus), free fatty acids are increased and promote the production of TG [20]. The resultant overproduction of TG enlarges VLDL associated with apolipoprotein C3, which delays the clearance of TG-rich lipoproteins (chylomicron and VLDL) from the blood [13]. Lipoprotein lipase is responsible for the TG hydrolysis at the luminal face of the capillary endothelium of each organ and thereby transforms chylomicrons and VLDL particles to TG-rich remnant lipoproteins and smaller lipoproteins particles [19]. TG-rich remnant lipoproteins can migrate into the subendothelial space and induce inflammation in the arterial wall to develop atherosclerosis [21]. TG-rich remnant lipoproteins include more cholesterols and TGs and are more proatherogenic than LDLs [21,22]. In the presence of abundant TG-rich lipoproteins, cholesterol ester transfer protein exchanges cholesterol ester in LDL and HDL for TG, which is hydrolyzed by hepatic lipase [13]. As a result, the size and density of LDL further decrease, turning into small dense LDLs, which are more atherogenic than large buoyant LDLs, while antiatherogenic HDL decreases [13]. Thus, elevated levels of postprandial TG indicate accumulation of proatherogenic TG-rich remnant lipoproteins (derived from chylomicrons and VLDLs) and small dense LDLs in the circulation, as well as a decrease in antiatherogenic HDLs, contributing to proatherogenic and procoagulant processes including inflammation, oxidative stress, and endothelial dysfunction [20,23,24]. The proatherogenic mechanisms involving TG are rather complicated and need to be further explored. However, therapies lowering TG levels could be one of important treatment strategies for suppressing atherosclerotic formation and preventing the progression of carotid artery stenosis.

There are some limitations to the present study. First, the present study included patients with well-controlled LDL-C levels but little control of elevated TG levels, which may have biased the results. In patients with carotid artery stenosis with elevated LDL-C levels, it is important to control LDL-C by administering statins as the first choice to prevent the carotid stenosis progression, and the enrolled patients had received the statin-based treatment at the time of the first visit to our hospital. However, patients with elevated TG levels have uncommonly undergone treatments such as fibrate administration, because the combination of statins and fibrates raises the risk of myopathy and rhabdomyolysis, and it remains unclear if TG-lowering therapy suppresses carotid stenosis progression. Based on the results of the present study, we plan to perform TG-lowering therapy using pemafibrate, which is considered to be safer in combination with statins [25]. Second, the present study did not consider local factors that may influence carotid stenosis progression, except for the initial degree of stenosis and SIR. Several studies have shown that carotid stenosis progression may be associated with certain local geometric and hemodynamic conditions [26], accounting for why carotid stenosis progression often occurs only on one side. Particularly, low wall shear stress and high oscillatory shear index in the carotid bulb may be a risk of carotid stenosis progression [27]. As assessment of patient-specific carotid geometry and local hemodynamics using computational fluid dynamics may enable stratification of the risk of carotid stenosis progression, it would be meaningful to study relationships between the local factors and non-fasting TG levels in carotid stenosis progression. Lastly, this was a retrospective study with a relatively small sample size. Therefore, the number and intervals of TG measurements were arbitrary and not protocolized. Further large-scale prospective studies are needed to demonstrate effects of changes in non-fasting TG levels and TG-lowering medications on carotid stenosis progression during the follow-up period.

## 4. Materials and Methods

### 4.1. Study Design

This was a single-center retrospective study. A total of consecutive eighty-eight patients (eighty-three males and five females) were recruited from patients with digital subtraction angiography (DSA) for carotid artery stenosis in our hospital from 1 January 2013 to 31 December 2019. The enrolled patients met the following inclusion criteria: aged 20 years or older at diagnosis, unilateral or bilateral atherosclerotic cervical internal carotid artery stenosis of ≥50% on the initial DSA according to the method used in the North American Symptomatic Carotid Endarterectomy Trial (NASCET), follow-up period of more than one year for less severe-side and/or worse-side carotid artery stenosis with no indication for carotid revascularization from the first DSA in our hospital, B-mode and color Doppler ultrasonography or magnetic resonance angiography of the carotid artery one or more times per year from the first DSA, and at least three blood samples taken during the follow-up period. The exclusion criteria were as follows: patients who underwent carotid artery stenting (CAS) or carotid endarterectomy (CEA) for carotid stenosis before 1 January 2013, carotid stenosis caused by dissection, and carotid artery stenosis with less than one-year follow-up from the first DSA to the last ultrasonography or CAS/CEA treatment. CAS or CEA was performed in accordance with the indication of carotid revascularization as described previously [28], and carotid stenosis treated with CAS or CEA was excluded from the study if the lesion had the indication of carotid revascularization at the first DSA and the period between the first DSA and CAS or CEA was less than one year; however, the non-operation-side carotid artery was included in the study if the artery was followed up for more than one year from the first DSA.

When patients first visited the outpatient department of our hospital, medical treatment was administered including antiplatelet agents and drugs for hypertension, dyslipidemia and diabetes mellitus if needed, according to the related Japanese guidelines. Routine laboratory data were assessed with non-fasting blood sampling to avoid patient burdens to fast, and the timing of the blood test was decided by the attending physician. The study was conducted to examine the relationships between carotid stenosis progression and each clinical variable, especially focusing on non-fasting TG.

The study was approved by the ethical committee of our institute and was performed according to the ethical standards of the responsible committee on human experimentation (institutional and national) and with the Helsinki Declaration of 1975, as revised in 2000 (World Medical Association Declaration of Helsinki 2000). For retrospective analyses, the committee waived the need for formal consent.

### 4.2. Clinical Data Collection

All clinical data were documented by experienced vascular neurosurgeons. The following data were gathered from medical records of 88 patients who underwent the first DSA during the period for extracranial (cervical) internal carotid artery stenosis: age, sex, body mass index, hypertension, diabetes mellitus, dyslipidemia, radiation induction, smoking, alcohol consumption, clinical symptoms (symptomatic or asymptomatic), degree of carotid artery stenosis (the NASCET method) on ultrasonography from the time of the first DSA diagnosis to the last follow-up or before CAS or CEA, the signal intensity ratio (SIR) of a carotid plaque on 3-dimensional T1-weighted gradient echo magnetic resonance images [29], atherosclerotic stenosis other than extracranial carotid artery (intracranial artery, subclavian artery, coronal artery, artery of lower extremities), drug profile (aspirin, clopidogrel, cilostazol, prasugrel, warfarin, direct oral anticoagulant, angiotensin receptor blocker, calcium channel blocker, β-blocker, statin, fibrate, eicosapentaenoic acid), non-fasting laboratory data at the first outpatient visit before the first DSA (total cholesterol, high-density lipoprotein cholesterol [HDL-C], non-HDL-C, LDL-C, TG, glucose, hemoglobin A1C), non-fasting TG values during the follow-up period, post-DSA treatment (CAS, CEA, medication only), and the duration of follow-up. When carotid artery stenosis was associated with ischemic events attributed to the lesion within 180 days before the first DSA, the lesion was defined as symptomatic. Outside of the period, carotid stenosis was considered asymptomatic.

Non-fasting blood sampling was measured within 10 h postprandially, and non-fasting hypertriglyceridemia was defined as non-fasting serum TG levels ≥ 175 mg/dL according to the European Atherosclerosis Society (EAS) and the European Federation of Clinical Chemistry and Laboratory Medicine joint consensus initiative [11]. Other clinical variables were defined as previously reported [6,28,30].

### 4.3. Analyses of Non-Fasting TG Values during the Follow-Up Period

Non-fasting TG levels were measured at least three times at different timing during the follow-up period. In addition to the maximum, minimum, and mean values of non-fasting TG during the follow-up period, the area of TG ≥ 175 mg/dL (area [TG ≥ 175], year-mg/dL) was calculated. The area [TG ≥ 175] was defined as the area determined by the time axis (year) and non-fasting TG values more than 175 mg/dL (i.e., TG values minus 175) during the follow-up period (Figure 2).

In this case, blood samples were taken five times during the follow-up period and the respective non-fasting TG values were 330 mg/dL on day 0, 169 mg/dL on day 268, 126 mg/dL on day 704, 188 mg/dL on day 732, and 330 mg/dL on day 907. The area [TG ≥ 175] in this case is 95.2 year-mg/dL, based on the following formula:



$\mathrm{Area}\;\left[\mathrm{TG}\geq 175\right]=\frac{1}{2}\left(330-175\;\langle \mathrm{mg}/\mathrm{dL}\rangle \right)\times \frac{268-0}{365}\;\langle \mathrm{year}\rangle \times \frac{330-175\;\langle \mathrm{mg}/\mathrm{dL}\rangle }{330-169\;\langle \mathrm{mg}/\mathrm{dL}\rangle }+\{\frac{1}{2}\left(188-175\;\langle \mathrm{mg}/\mathrm{dL}\rangle \right)\times \frac{732-704}{365}\langle \mathrm{year}\rangle \times \frac{188-175\;\langle \mathrm{mg}/\mathrm{dL}\rangle }{188-126\;\langle \mathrm{mg}/\mathrm{dL}\rangle }+\frac{1}{2}\left(\left[188-175\;\langle \mathrm{mg}/\mathrm{dL}\rangle ]+[330-175\;\langle \mathrm{mg}/\mathrm{dL}\rangle \right]\right)\times \frac{907-732}{365}\;\langle \mathrm{year}\rangle \}=95.2$



### 4.4. Progression of Carotid Artery Stenosis

Among 176 enrolled carotid arteries in 88 patients, one carotid artery was excluded because of complete occlusion at the initial DSA. In addition, 64 worse-side carotid arteries had CAS or CEA following the first DSA, and were therefore excluded. Thus, the remaining 111 carotid arteries on the less severe stenosis side and worse stenosis side with no indication for carotid revascularization were analyzed. In the analysis, carotid artery stenosis was divided into two groups, progression and non-progression groups. The progression of carotid artery stenosis was diagnosed only using ultrasonography as previously described [6]. The progression was defined as ≥10% increase (the NASCET method) in the degree of carotid stenosis on ultrasonography during the follow-up period compared with that on ultrasonography at the initial DSA. Clinical variables were compared between the two groups.

### 4.5. Statistical Analysis

All data were analyzed using the SPSS software version 25.0 (IBM, Armonk, NY, USA). Categorical variables were reported as a proportion and were analyzed using chi-square or Fisher’s exact tests, as appropriate. Continuous variables were reported as a mean ± standard deviation (SD) or a median (interquartile range) and were compared using unpaired t or Wilcoxon rank-sum tests between the two groups, as appropriate. The impact of each variable on carotid stenosis progression was determined by multivariate logistic regression analyses using the dichotomous status (progression or non-progression) as the dependent variable. Variables were selected if univariate association was *p* < 0.05 on univariate analyses, although only the variable with the smallest probability value was used as a candidate variable among similar clinical variables that were intercorrelated (Pearson’s or Spearman’s correlation coefficient [r] ≥0.4). Adjusted odds ratios (ORs) with 95% confidence intervals (CIs) were calculated, and the independence of variables was tested using the likelihood ratio test on reduced models. The receiver operating characteristic (ROC) curves with area under the curve (AUC) for the area [TG ≥ 175] were calculated in the reference population by carotid stenosis progression. *p* values less than 0.05 were considered significant.

## 5. Conclusions

Symptomatic carotid stenosis and the area [TG ≥ 175] were independent risk factors for subsequent progression of carotid artery stenosis in patients with ≥50% carotid artery stenosis at least on one side. The findings suggest the importance of continuous monitoring of non-fasting TG levels to predict the progression of carotid artery stenosis especially in symptomatic cases. Further studies are warranted to explore the impact of non-fasting TG burden during a follow-up on carotid stenosis progression and to investigate the possibility of a TG-lowering therapy to prevent it.

## Figures and Tables

**Figure 1 ijms-23-09197-f001:**
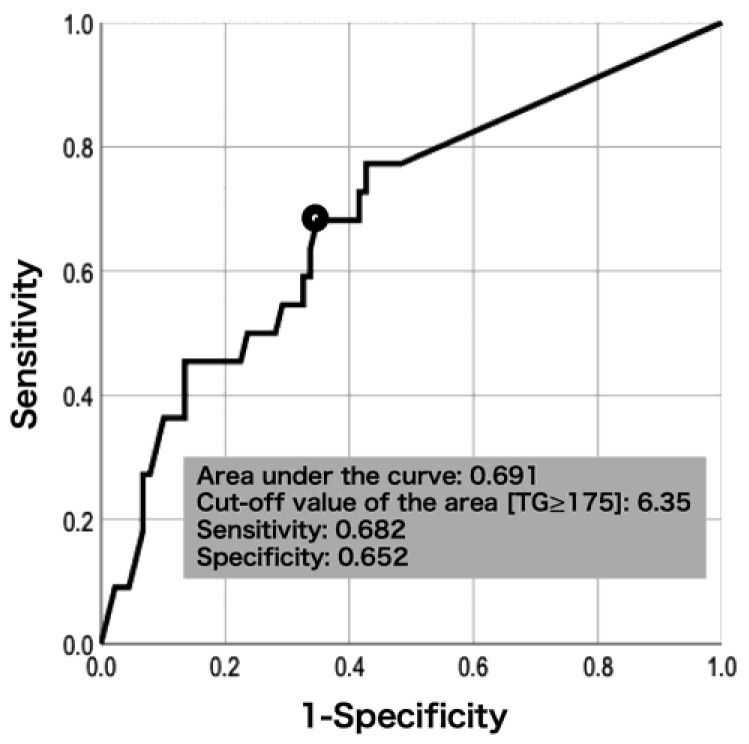
Receiver operating characteristic curve for the area [TG ≥ 175] in the reference population by carotid stenosis progression.

**Figure 2 ijms-23-09197-f002:**
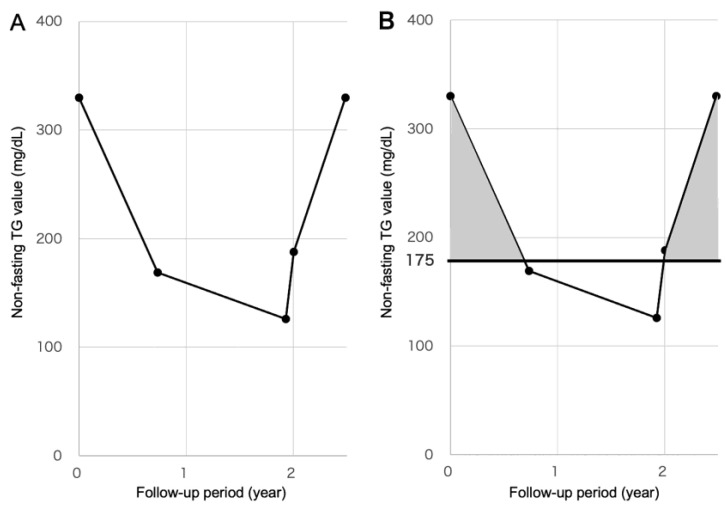
Calculation of the area [TG ≥ 175], which indicates cumulative non-fasting TG values ≥ 175 mg/dL during a follow-up period. (**A**) Non-fasting TG values measured during a follow-up period are shown on the line graph. (**B**) The area of non-fasting TG values ≥ 175 mg/dL during the follow-up period is calculated and the value is defined as the area [TG ≥ 175] (year-mg/dL; *gray area*).

**Table 1 ijms-23-09197-t001:** Comparisons of baseline clinical characteristics and non-fasting laboratory data at the initial diagnosis between the subsequent carotid stenosis progression and non-progressions groups.

Variable	Progression (*n* = 22)	Non-Progression (*n* = 89)	*p* Value	Odds Ratio
Age (y)	66.6 ± 10.5	71.6 ± 6.7	0.044 ^a^	
Male	21	85	1 ^c^	1.012
Body mass index (kg/m^2^)	23.3 ± 3.5	22.7 ± 3.4	0.426 ^a^	
Smoking	18	72	0.921 ^c^	1.063
Alcohol consumption	12	47	0.884 ^b^	1.072
Carotid stenosis				
Symptomatic	9	9	<0.001 ^b^	6.154
Degree of stenosis (%)	32.6 ± 28.4	20.5 ± 31.2	0.088 ^a^	
Signal intensity ratio	1.49 ± 0.39	1.43 ± 0.58	0.264 ^a^	
Radiation induced	3	3	0.091 ^c^	4.526
Past medical history				
Hypertension	18	68	0.777 ^c^	1.390
Diabetes mellitus	17	54	0.146 ^b^	2.204
Hyperlipidemia	11	47	0.813 ^b^	0.894
Chronic kidney disease	4	11	0.492 ^c^	1.576
Other atherosclerotic stenosis				
Intracranial artery	3	10	0.719 ^c^	1.247
Subclavian artery	1	3	1 ^c^	1.365
Coronal artery	9	43	0.533 ^b^	0.741
Artery of lower extremities	2	10	1 ^c^	0.790
Medication				
Aspirin	10	49	0.361 ^b^	0.646
Clopidogrel	17	53	0.153 ^b^	2.181
Cilostazol	11	41	0.809 ^b^	1.122
Prasugrel	0	8	0.355 ^c^	0
Warfarin	1	2	0.495 ^c^	2.024
Direct oral anticoagulant	2	5	0.627 ^c^	1.640
Angiotensin receptor blocker	7	16	0.168 ^b^	2.071
Calcium channel blocker	5	37	0.088 ^b^	0.397
Statin	16	57	0.521 ^b^	1.404
Fibrate	0	1	1 ^c^	0
Eicosapentaenoic acid	0	3	1 ^c^	0
Non-fasting laboratory data				
Total cholesterol (mg/dL)	182.3 ± 43.9	171.6 ± 35.9	0.238 ^a^	
HDL-C (mg/dL)	46.7 ± 11.2	53.4 ± 15.0	0.081 ^a^	
Non-HDL-C (mg/dL)	137.9 ± 44.8	121.4 ± 37.1	0.093 ^a^	
LDL-C (mg/dL)	104.8 ± 39.4	98.4 ± 29.1	0.409 ^a^	
TG (mg/dL)	230.1 ± 118.1	151.7 ± 82.4	<0.001 ^a^	
Hypertriglyceridemia	14	28	0.005 ^b^	3.810
Glucose (mg/dL)	157.3 ± 69.4	132.7 ± 55.2	0.087 ^a^	
Hemoglobin A1C (%)	6.99 ± 1.16	6.73 ± 1.15	0.370 ^a^	
Follow-up period (days)	1315.6 ± 988.6	1153.0 ± 679.3	0.472 ^a^	

Values are a mean ± standard deviation or the number of cases. Continuous and categorical variables are compared using unpaired t ^a^, chi-square ^b^ or Fisher’s exact ^c^ tests, as appropriate. Abbreviations: HDL-C, high-density lipoprotein cholesterol; LDL-C, low-density lipoprotein cholesterol; TG, triglyceride.

**Table 2 ijms-23-09197-t002:** Comparisons of non-fasting triglyceride (TG) profile during the follow-up period between the carotid stenosis progression and non-progressions groups.

Variable	Progression (*n* = 22)	Non-Progression (*n* = 89)	*p* Value	Odds Ratio
Maximum TG (mg/dL)	330.2 ± 370.6	213.8 ± 206.9	0.050 ^a^	
Maximum TG ≥ 175 mg/dL	17	46	0.030 ^c^	3.18
Minimum TG (mg/dL)	116.5 ± 47.8	97.0 ± 46.9	0.085 ^a^	
Minimum TG ≥ 175 mg/dL	2	6	0.657 ^d^	1.38
Mean TG (mg/dL)	206.5 ± 135.1	146.9 ± 88.9	0.013 ^a^	
Mean TG ≥ 175 mg/dL	10	21	0.041 ^c^	2.69
Area [TG ≥ 175] (year-mg/dL)	27.7 (2.6−411.6)	0 (0−28.0)	0.004 ^b^	
Area [TG ≥ 175] ≥ 6.35 year-mg/dL	15	30	0.003 ^c^	4.21

Values are a mean ± standard deviation, a median (interquartile range), or the number of cases. Continuous and categorical variables are compared using unpaired t ^a^, Wilcoxon rank-sum ^b^, chi-square ^c^ or Fisher’s exact ^d^ tests, as appropriate.

**Table 3 ijms-23-09197-t003:** Multivariate logistic regression analyses of clinical variables for carotid stenosis progression.

Variable	Odds Ratio	95% Confidence Interval	*p* Value
Symptomatic case	7.167	2.168–23.700	0.001
Area [TG ≥ 175] ≥ 6.35 year-mg/dL	4.827	1.626–14.329	0.005

## Data Availability

Data from this study will be made available to qualified investigators upon reasonable inquiry.

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
