# Peer review of "Non-Fasting Hypertriglyceridemia Burden as a Residual Risk of the Progression of Carotid Artery Stenosis"

_ijms, 2022, doi:10.3390/ijms23169197_

Round 1

Reviewer 1 Report

1.      Please make a statement of the limitations on your study. For example, the small sample size, retrospective study characteristics, etc.

2.      What’s the indication of fibrate usage? According to the results, 17 of 22 in the progression group and 46 of 89 in the non-progression group were identified with hypertriglyceridemia. However, only 1 patient received fibrate therapy. Please give some explanations.

3.      The patients in progression group were significantly younger than the ones in non-progression group, please make some discussion.

4.      As the symptomatic carotid stenosis is an independent risk factor for the progression, I wonder if the study could reveal the correlation between non-fasting TG level and symptomatic carotid stenosis?

Author Response

Thank you so much for your insightful suggestions. We have revised our paper in accordance with the suggestions offered, and the revisions are red-characterized.

  1. Please make a statement of the limitations on your study. For example, the small sample size, retrospective study characteristics, etc.

Answer:

According to your suggestions, we added a statement of the limitations of the present study to the last paragraph in the Discussion section in page 12.

  1. What’s the indication of fibrate usage? According to the results, 17 of 22 in the progression group and 46 of 89 in the non-progression group were identified with hypertriglyceridemia. However, only 1 patient received fibrate therapy. Please give some explanations.

Answer:

In this study, fibrates were generally not administered. According to your suggestions, we described the reason as the 1st limitation of this study in page 12 as follows: “First, the present study included patients with well-controlled LDL-C levels but little control of elevated TG levels, which may have biased the results. In patients with carotid artery stenosis with elevated LDL-C levels, it is important to control LDL-C by administering statins as the first choice to prevent the carotid stenosis progression, and the enrolled patients had received the statin-based treatment at the time of the first visit to our hospital. However, patients with elevated TG levels have uncommonly undergone treatments such as fibrate administration, because the combination of statins and fibrates raises the risk of myopathy and rhabdomyolysis, and it remains unclear if TG-lowering therapy suppresses carotid stenosis progression. Based on the results of the present study, we plan to perform TG-lowering therapy using pemafibrate, which is considered to be safer in combination with statins [28].”

  1. The patients in progression group were significantly younger than the ones in non-progression group, please make some discussion.

Answer:

Thank you for your suggestions. We think that the findings probably reflect the results of younger patients preferring diets that raise TG values. According to your suggestions, we added discussion in the second paragraph in the Discussion section in page 10 as follows: “It is well known that advanced age itself is related with pathogenic mechanisms of the development of carotid artery atherosclerosis in multiple ways [11]. In the present study, however, patients in the carotid stenosis progression group were significantly younger than those in the non-progression group on univariate analyses. The discrepancy can be explained by the finding that age was significantly correlated with non-fasting TG values. It has been documented that lifestyle factors including acute and habitual diet influence non-fasting TG, and higher TG values were sometimes reported in younger patients [12,13]. The findings in the present study probably reflect the results of younger patients preferring diets that raise TG values.”

  1. As the symptomatic carotid stenosis is an independent risk factor for the progression, I wonder if the study could reveal the correlation between non-fasting TG level and symptomatic carotid stenosis?

Answer:

Thank you for your suggestions. In our study, symptomatic carotid stenosis was not significantly correlated with non-fasting TG-related variables including initial TG levels (r=0.05), maximum TG levels (r=0.02), mean TG levels (r=0.03), area [TG≥175] (r=0.05) and the incidence of area [TG≥175] ≥6.35 (r=0.04). According to your suggestions, we added the following sentence to the subsection 3.3. in the Results section in page 9: “Symptomatic carotid stenosis was not significantly correlated with non-fasting TG-related variables including the incidence of area [TG≥175] ≥6.35 year-mg/dL (r=0.04).”

Reviewer 2 Report

Based on a retrospective data analysis of 88 patients with carotid artery stenosis, the authors suggest that higher levels of non-fasting TG in blood are associated with further progression of stenosis.  This conclusion is only correlative as no mechanistic approaches were taken.  Overall, the paper is easy to follow although there are some points that need to be addressed.  Please see below.

1. Abstract:  Line 5.  "The area was calculated.....to evaluate non-fasting TG burden."  What area?  This statement, as it is written, is incomprehensible.  If the authors wish to use the term "area", it must be clearly defined before using it.

2. When calculating the "area", it is clear from Fig. 1 that this value is a function of how many times blood sampls were collected from a patient (i.e. number of data points) and the time interval between two successive samplings.  In the paper, these data were glossed over.  It may help readers if all the TG meanurement data are shown as supplement.

3. TG measurement values are a systemic factor, but often, restenosis is detected only on one side.  This suggests the presence of some local factor as well.  Please consider discussing this.

Author Response

Thank you so much for your insightful suggestions. We have revised our paper in accordance with the suggestions offered, and the revisions are red-characterized.

Based on a retrospective data analysis of 88 patients with carotid artery stenosis, the authors suggest that higher levels of non-fasting TG in blood are associated with further progression of stenosis.  This conclusion is only correlative as no mechanistic approaches were taken.  Overall, the paper is easy to follow although there are some points that need to be addressed.  Please see below.

  1. Abstract: Line 5. "The area was calculated.....to evaluate non-fasting TG burden." What area? This statement, as it is written, is incomprehensible. If the authors wish to use the term "area", it must be clearly defined before using it.

Answer:

According to your suggestions, we revised the sentence in the Abstract section in page 1 as follows: “To evaluate non-fasting TG burden, a new parameter area [TG≥175] was calculated by integrating non-fasting TG values ≥175mg/dL (i.e., TG values minus 175) with the measurement intervals (year).”

  1. When calculating the "area", it is clear from Fig. 1 that this value is a function of how many times blood samples were collected from a patient (i.e. number of data points) and the time interval between two successive samplings. In the paper, these data were glossed over. It may help readers if all the TG measurement data are shown as supplement.

Answer:

Thank you for your suggestions. According to your suggestions, we have shown all the TG measurement data as supplementary tables (Tables S1 and S2).

  1. TG measurement values are a systemic factor, but often, restenosis is detected only on one side. This suggests the presence of some local factor as well.  Please consider discussing this.

Answer:

Thank you for your suggestions. We agree with your opinion. According to your suggestions, we added the discussion as the 2nd limitation of this study in page 12 as follows: “Second, the present study did not consider local factors that may influence carotid stenosis progression, except for the initial degree of stenosis and SIR. Several studies have shown that carotid stenosis progression may be associated with certain local geometric and hemodynamic conditions [29], accounting for why carotid stenosis progression often occurs only on one side. Particularly, low wall shear stress and high oscillatory shear index in the carotid bulb may be a risk of carotid stenosis progression [30]. As assessment of patient-specific carotid geometry and local hemodynamics using computational fluid dynamics may enable stratification of the risk of carotid stenosis progression, it would be meaningful to study relationships between the local factors and non-fasting TG levels in carotid stenosis progression.”

References

[29] Lee SW, Antiga L, Spence JD, Steinman DA. Geometry of the carotid bifurcation predicts its exposure to disturbed flow. Stroke2008;39:2341-2347. http://doi.org/10.1161/ STROKEAHA.107.510644.

[30] Markl, M. Wegent F, Zech T, Bauer S, Strecker C, Schumacher M, Weiller C, Hennig J, Harloff A. In vivo wall shear stress distribution in the carotid artery: effect of bifurcation geometry, internal carotid artery stenosis, and recanalization therapy. Circ Cardiovasc Imaging 2010;3:647-655. http://doi.org/10.1161/CIRCIMAGING.110.958504.

Round 2

Reviewer 1 Report

NA